# Application of Lean 6s Methodology in an Engineering Education Environment during the SARS-CoV-2 Pandemic

**DOI:** 10.3390/ijerph17249407

**Published:** 2020-12-15

**Authors:** Mariano Jiménez, Luis Romero, Jon Fernández, María del Mar Espinosa, Manuel Domínguez

**Affiliations:** 1Mechanical Engineering Department–ICAI, Comillas University, 28015 Madrid, Spain; mjimenez@icai.comillas.edu; 2Design Engineering Area, Department of Construction and Fabrication Engineering–UNED, 28040 Madrid, Spain; jfernande5052@alumno.uned.es (J.F.); mespinosa@ind.uned.es (M.d.M.E.); mdominguez@ind.uned.es (M.D.)

**Keywords:** lean, 6S, PDCA, safety, SARS-CoV-2

## Abstract

In this work, the application of the Lean 6S methodology is exposed, which includes the Safety-Security activity in response to the demands caused by the epidemiological situation due to exposure to SARS-CoV-2, as well as its implementation through a standardized process in n higher education environment in the engineering field. The application of methodologies based on lean principles in the organizational system of an educational institution, causes an impact on the demands of organizational efficiency, where innovation and continuous improvement mark the path to success. The Lean 6S methodology, based on the development of six phases, guarantees, thanks to the impact of all its phases and especially of three of them: cleaning, standardize and safety, the control of the health risk against SARS-CoV-2. This guarantee is achieved through the permanent review of safety in the workplace. The areas of selected implementation to verify the effect have been the essential spaces for the development of the teaching activity: center accesses, learning rooms and practical laboratories. The laboratories are adapted to the security and organization conditions that are required in the regulations required by the Occupational Risk Prevention Services against exposure to SARS-CoV-2, since the appropriate protective equipment for the risk level is reviewed, the ordering of the workstations, the class attendance through the shifts organization and the rearrangement of the common places where the maintenance of a minimum interpersonal safety distance between the teaching staff, auxiliary services and students is guaranteed. The effort of the teaching staff in terms of following the established rules is notably increased. To balance this dedication, it is necessary to increase and rely on auxiliary personnel who guarantee rules compliance control in different spaces than the classroom and the laboratory.

## 1. Introduction

In this work, the application of the Lean 6S methodology is exposed, which includes the Safety-Security activity in response to the demands caused by the epidemiological situation due to exposure to SARS-CoV-2, as well as its implementation through a standardized process in a higher education environment in the engineering field. This study was carried out in an engineering school in Madrid (Spain) and under the operational criteria established by the health and labor authorities of the Government of Spain synchronized with the European Commission through the European Center for Disease Prevention and Control (ECDC).

Government organizations responsible of health management have generated “Action procedures for Occupational Risk Prevention Services against exposure to SARS-CoV-2” [1]. In them, it is up to the companies to assess the exposure risk in which workers may find themselves in each of the activities carried out and to follow the recommendations issued by the prevention service on the matter.

Given the possible exposure scenarios to the SARS-CoV-2, educational institutions have had to, among other activities, review the appropriate protective equipment for the risk level, the ordering of workstations, attendance to class by organizing shifts and the reorganization of common places where the maintenance of a minimum interpersonal safety distance is guaranteed between faculty members, auxiliary tech staff and students [2,3,4,5].

The application of methodologies based on lean principles in the organizational system of an educational institution, causes an impact on the demands of organizational efficiency, where innovation and continuous improvement set the path to success [6]. The Lean 6S methodology, based on the development of six phases, guarantees the control of the health risk against SARS-CoV-2 [7,8,9,10]. This guarantee is achieved through the permanent review of safety in the workplace. The areas of implementation selected to verify the effect have been the essential spaces for the development of the teaching activity: center access, learning rooms and practical laboratories.

The word Lean adapted to a process seeks to maximize value creation and minimize the amount of useless, expendable, or negative resources. The 5S are initially based on Japanese acronyms of seiri (organization), seiton (order), seiso (cleaning), seiketsu (standardization) and shitsuke (sustain). A framework of applying 5S within a business (as opposed to a personal philosophy of way of life) was first formalized in the early 1980s by Takashi Osada. The practice of 5S aims to embed the values of organization, neatness, cleaning, standardization and discipline into the workplace [11].

The Lean 5S are universal; they can be applied in all types of companies and organizations, both in workshops, laboratories and in offices, even in those that apparently are sufficiently organized and clean.

You can always avoid inefficiencies, avoid displacements, and eliminate waste of time and space. The initial Lean 5S methodology is implemented in many companies and institutions and its effects have generated an increase in competitiveness through its influence on continuous improvement caused by order and organization in the workplace [12].

The natural evolution of the 5S and the need to respond to a security requirement in the methodology made the inclusion of a sixth S mandatory, safety (sekyuriti) (6S). With this expansion, the reduction and elimination of the different risks that may exist in a workplace is achieved, through compliance with current regulations, both in the prevention of occupational risks and in the safety of the industrial resources used [13].

Faced with the situation caused by the SARS-CoV-2, the Lean 6S methodology allows the exhaustive review of each process and work area, in order to guarantee safety in all work areas that are deliberately or unintentionally exposed to SARS-CoV-2 [14,15]. The main objectives of this application in an educational center would be [16]:Application assurance of the safety regulations for SARS-CoV-2.Application assurance of the certification required in the Personal Protective Equipment (PPE).Guarantee of minimum contagion in an environment with implicit security.

It is important to note that the Lean 5S methodology has been successfully applied in many different fields in order to combat the effects associated with the SAR-CoV-2 pandemic [14,17,18,19,20,21,22,23].

## 2. Development

The starting data for the development of this proposal are conditioned to the initial investigations on the virus transmission:The transmission route between humans is considered similar to that described for other coronaviruses through the secretions of infected people, mainly by direct contact with respiratory droplets larger than 5 microns (capable of being transmitted at distances of up to 2 m) and contaminated hands or parts with these secretions followed by contact with the mucosa of the mouth, nose and eyes [24]. SARS-CoV-2 has been detected in nasopharyngeal secretions, including saliva [25].The permanence of viable SARS-CoV-2 on copper, cardboard, stainless steel and plastic surfaces was 4, 24, 48 and 72 h, respectively at 21–23 °C and with 40% relative humidity [26]. In another study, at 22 °C and 60% humidity, the virus was no longer detected after 3 h on a paper surface (printing or tissue paper), after 1 to 2 days on wood, clothing or glass and more than 4 days on stainless steel, plastic, money bills and surgical masks [27].

In order to establish the required preventive measures, Table 1 shows the different exposure scenarios to SARS-CoV-2 in which the workers and students of the center can be found.

In order to know in what situation we are in each of the described scenarios above, Figure 1 defines the virus transmission risk in different situations [28].

The response to these scenarios requires prevention activities development with their technical application procedures, which include preventive recommendations and guidelines that must be reflected in a Contingency Plan against SARS-CoV-2.

This practical, clear and real Plan must be prepared by the Directorate/Management with the support of those responsible and intermediate positions and the representatives of the workers, and must:Prevent infection caused in the areas of the center through preventive measures.Respond efficiently to the appearance of cases among the personnel involved.Act in the event of contact with infected persons.Plan the progressive reincorporation of staff according to the priority level of each service or activity and social conditions: family situation, the proximity of the home to the workplace, distance resources in the workspace and the need for non-remote tasks.

Like any of the 6S activities, the Contingency Plan must be subject to a permanent audit to guarantee the application and monitoring of the measures.

Safe work procedures for SARS-CoV-2 must guarantee the use of collective and individual protection equipment in the educational center, and good practice procedures, without forgetting the safe handling of laboratory practice equipment.

These measures should be incorporated into any previous working protocol with students. The following procedures are considered essential:Procedure for organizing access to work or study.Procedure for using the resources of the job or study.Procedures of facilities cleaning and decontamination, equipment and material used in the development of teaching activity.Waste management procedure that may contain SARS-CoV-2 in personal and occupational safety protection PPE.Procedure for the security conditions of the center. Existence of individual and collective protection equipment, and of ensuring the safety distance.Procurement procedure to guarantee the necessary quantity and supply of PPEs and hygiene, cleaning and decontamination material.Procedure for security and cleaning services provided by companies.Procedure for establishing communication and information channels with all the personnel involved.Procedure for raising awareness of the exceptional situation to integrate preventive activity at all levels of the organization and encourage staff participation.

The study was conducted in accordance with the Declaration of Helsinki, and the protocol was approved by the Ethics Committee of Comillas University (approval number 49–20).

## 3. Methodology: Adaptation of the Lean 6S Methodology

As a complement to these mandatory procedures and measures to guarantee the safety and health of the staff, the use of the Lean 6S methodology generates a habit based on commitment, participation and order in the center.

In Figure 2 is shown that the Lean 6S procedure is based on six phases [13]:(a)Organization (Seiri).(b)Order (Seiton).(c)Cleaning (Seiso).(d)Safety (Sekyuriti).(e)Standardize (Seiketsu).(f)Sustain (Shitsuke).

The impact of each of the phases against SARS-CoV-2 is shown in Table 2.

The involvement of these six phases in the procedures described above, should be carried out in its entirety and in a synchronized way. Three of the six phases may have a greater impact on SARS-CoV-2’s contagion demands, but the success of the implementation of the methodology requires a rigorous order of implementation. The first four phases are operational, the five-phase maintains the state reached with the previous phases and the six-phase helps us to work for continuous improvement.

The procedure followed to implement the methodology Lean 6S was as follows:Obtaining the commitment from the management of the Center that sets the depth and duration of the project.Definition of the work team:
A team of teaching and non-teaching staff who take part in the involved laboratories.A guide, which provides documentation, training and resources to the team.
Implementation in a reference area (pilot) to thoroughly learn the methodology and develop an enhancement that serves as an example.Implementation elsewhere in access, corridors, classrooms and laboratories (generalization).

The developed functions for each of the participants provided in the process implementation are exposed in Table 3, while Table 4 shows the information of the students who have participated in the study.

The stages that have been followed to carry out the 6S process implantation process have followed the execution order showed in Table 5.

The 6S implementation process is carried out following the Plan–Do–Check-Action (PDCA) cycle in each of the phases. It begins with the preparation of the phase, then data will be taken from the work area, analyzed to establish an improvement plan and finally rules of action are defined.

The implementation process of the Lean 6S methodology in the different work areas requires detailed procedures that avoid conceptual confusion among the participants. Figure 3 shows the procedure to follow in phase 4 to achieve the required level of security.

The use of indicators that show the evolution of the implementation process provides a real situation of the improvement achieved.

Some of these indicators need to be customized over time to really show what is happening in each of the implementation areas. The main indicators that have been proposed are:The degree of compliance with the established program.Laboratory practice preparation time.Theory class preparation time.Time of access to the job place.Time lost for adaptation to security conditions.Risk situations due to improper use of protective equipment (PPE).Maintenance and replacement costs.Identification of security anomalies.Rate of anomalies susceptible to contagion.The number of positive cases confirmed with SARS-CoV-2.Suspicion of cases due to symptoms compatible with SARS-CoV-2.Close contact with confirmed positive with SARS-CoV-2.

### 3.1. PHASE 1–Organization (Seiri)

The first phase allows you to search and identify all the unnecessary elements that can cause anomalies or situations of insecurity. These elements are marked with specific labels and their exact location. Subsequently, the action established to transfer or relocate all the remaining elements is decided.

The characteristics of this phase are:Advantages: elimination of obsolete and duplicate objects; use of space; reduced sense of disorganization.Obstacles: Confusing definitions of necessary and unnecessary; Accumulation of items pending classification that take up space; Waste of time deciding what is necessary.Indicators: Number of unnecessary; m^2^ of soil released.

### 3.2. PHASE 2–Order (Seiton)

In this phase, the elements necessary for the correct performance of work activities (teaching) in each area must be identified and placed. A list of these elements is created, with a description of their use and the amount necessary to operate correctly.

Phase characteristics:Advantages: materials are easily found; reduction of the movement of personnel and materials; comfort and safety to pick up materials; stock reduction; increased security.Obstacles: free spaces that become occupied by other objects; poorly planned allocation of locations.Indicators: number of elements outside the assigned place; number of unidentified materials, elements and areas.

### 3.3. PHASE 3–Cleaning (Seiso)

The main objective is to eliminate dirt and dirty sources that are active and can enhance virus transmission. The workplace must always be in optimal working conditions, clean and disinfected. Previously, it is necessary to indicate some concepts that condition the interpretation of the procedures [29].

First, regarding the processes related to cleaning:Decontamination: process in which microorganisms present on contaminated or suspected contaminated surfaces are eliminated.Disinfection: physical or chemical process applied to contaminated materials or to skin surfaces, and used to kill the microorganisms present, but not necessarily their spores.Sterilization: any process that destroys or eliminates any kind of microorganism, whether it is in the vegetative phase or in the form of spores.

Second, regarding decontamination agents:Biocide: general term used for any agent that kills viruses, unicellular and multicellular organisms.Chemical germicide: chemical agent or mixture of chemical agents used to kill microorganisms.Virucidal: substance or drug capable of destroying or inactivating viruses.Disinfectant: chemical agent or mixture of chemical agents used to kill microorganisms, but not necessarily their spores.Antiseptic: substance that inhibits the growth and development of microorganisms without necessarily causing their death.

In accordance with the recommendation to disinfect surfaces to stop virus human transmission, virucidal products authorized by health authorities should be used. In Spain, the list of virucidal products that have demonstrated efficacy against viruses has been published, according to the UNE-EN 14476: 2014 + A2: 2020 standard (Antiseptics and chemical disinfectants. Quantitative suspension test for the evaluation of virucidal activity in medicine. Test method and requirements -Phase 2/Stage 1-).

The main characteristics of this phase are:(a)Advantages: improved safety and health risks elimination; due to the fact that dirty sources (sharp material) are eliminated and complex cleaning activities of difficult access areas are avoided; reduction of interruptions due to the cleaning need; optimization of waste management; visibility of anomalies and maintenance improvements.(b)Obstacles: complexity when evaluating the present cleaning degree; damages for considering it an outside activity; resignation with dirty sources.(c)Indicators: number of dirty sources; number of damaged materials; number of difficult places; execution time.(d)Recommendations:
Use stock control cards/keyrings on all material, including description, product reference and max./min. necessary.Indicate location of cleaning elements. Mark mandatory cleaning sources.The dirt is not only stains and dust, it is also the accumulation of documentation, materials and some unnecessary elements.

Against SARS-CoV-2, hand hygiene is a vital measure of infection prevention and control; it is considered that in the educational center it should be carried out on the following occasions:After blowing your nose, coughing, or sneezing.After going to the bathroom.After visiting a common access space.After touching uncontrolled surfaces in the classroom or laboratory environment: A plastic container, notebooks, partner’s pen, control equipment, money, etc.Whenever it can be in contact with blood, body fluids, non-intact skin and mucous membranes.Before and after eating.Before putting on the personal protective equipment and after its removal.

In the case of laboratories test, it is necessary to consider:(a)Having used gloves does not exempt from performing proper hand hygiene after removal and nails should be kept short and neat, avoiding wearing rings, bracelets, wristwatches or other ornaments.(b)If hands are visibly clean, hand hygiene will be done with products with a minimum of 70% alcoholic base; on the other hand, if hands are visibly dirty or stained with fluids, will do with antiseptic soap and water.(c)Personal protective equipment PPE and equipment or technical resources must be decontaminated by any of the following procedures:
Methods of disinfection and sterilization with chemical germicides.Saturated steam sterilization (autoclaves).Use of dry heat.Use of ultraviolet radiation.Laboratory extraction air filtration.


The basic procedure of common surfaces decontamination in the educational center is carried out with chemical germicides, and consists of the following stages:Pre-cleaning.Application of the disinfectant (spray, cloth, etc.).Wait time for action.Disinfectant washing, if appropriate depending on the used disinfectant.

Five methods of decontamination, scrubbing, spraying, fumigation, immersion and gas-phase diffusion [30] are commonly recommended. In a general educational center such as the one indicated in this study, the use of:Scrubbing. It can be applied in any type of laboratory and in other areas of common use such as access corridors, elevators, hall, etc. A previous cleaning must be carried out using surfactant compounds that remove organic matter from the surfaces. The operation depends on the surface to be cleaned.Spraying. It can be applied in any type of laboratory and in other areas of common use such as access corridors, elevators, hall, etc. It is used to clean walls and ceilings using sprayers or pressure fumigators to diffuse the germicide over the surface to be disinfected. This method also uses special mops and mops that do not drip and do not shed textile particles.

In the case of laboratories test and depending on the equipment used (machine tools, measuring equipment, etc.) the possibility of using ultraviolet radiation and the revision of the filtration and renewal of the laboratory’s extraction air should be analyzed. Ultraviolet light has limited reliability as its effect only occurs if surfaces are very clean, there is relative humidity ≥70%, there are no shadows or distorting reflections and the distance from the emitting lamp to the surface is less than 90 cm.

The renewal and filtration of air in spaces where laboratory teaching practices take place, there is no direct exposure to biological agents and the inactivation of the extracted air is not mandatory. In the centers there is usually a centralized ventilation system with recirculation of air, it must be ensured that the air supplied is 20 L/second and person (or equivalent to maintain a maximum CO_2_ concentration of 350 ppm), properly treated depending on the quality of outside air.

### 3.4. PHASE 4–Safety (Sekyuriti)

This is the phase that focuses on safety, and the objective is to reduce occupational risks for workers in the work area and to ensure that the work area complies with current regulations for the prevention of contagion risks, connected to work and technical resources use rules. As in the rest of the phases, it is necessary, and more in view of the novelty that the SARS-CoV-2 effect supposes, the definition of elements, tasks and a flow chart to sequence the necessary implementation activities that guarantee safety. The use of checklists is essential to analyze the safety level of the work area, detect potential risks, identify them and apply rules so that the personnel involved can work in an adequate and safe environment. A flow chart was designed where reference is made to three different checklists:Checklist 1 detects the exposure scenario in which the resources are found to establish required preventive measures.The objective of checklist 2 is to guide technicians when adapting machines that do not have the CE marking to the requirements established in current legislation, directive 2006/42/CE.Checklist 3 informs technicians about all the necessary PPE related to the job in question. To achieve this, a table was created that establishes the relationship between the different parts of the body and the corresponding PPE according to Regulation (EU) 2016/425 Of The European Parliament and Of The Council, of March 9, 2016, regarding individual protection equipment.

Within the flow diagram (Figure 3), the bases of the implementation process of the Supervise Security phase in the application area are established. Following the steps in this flow chart, the security of the selected elements is analyzed, checking the level of risk of those involved, the CE marking of the laboratory practice resources and the necessary PPEs according to the risk level and the resource.

### 3.5. PHASE 5–Standardize (SEIKETSU)

The standardization phase aims to eliminate possible situations in which there are values outside the established limits, so that everything is always in a regular situation and in its designated position. Furthermore, by means of signage, it was sought to simplify all the basic tasks of access, location and mobility so that students do not waste time with doubts.

This is done using several horizontal and vertical marking systems (colored tapes, prohibition, warning, obligation and information signs, etc.). Examples of signaling are shown in Figure 4 and Figure 5 [7].

Phase characteristics:Advantages: it facilitates the order and cleaning maintenance, irregular situations are detected immediately, active knowledge of functions and stock levels is generated, and an increase in control and security is achieved.Obstacles: difficulty to establish the maximum and minimum quantities, technical difficulty in implementing signs on machines and equipment, as well as horizontal signposting surfaces.Indicators: relationship between the number of marked points and the number of necessary points.

### 3.6. PHASE 6–Sustain (Shitsuke)

The sixth and last phase is based on carrying out periodic analysis of the implantation area, to ensure that all the changes introduced with the 6S methodology are fulfilled.

In this way, it is possible to maintain and increase the benefits obtained during the five phases already implemented, while establishing a routine environment. If the new processes and established changes are not accepted or respected, the whole implementation will have been in vain. It is not only about auditing the pilot area, but also about finding potential improvement mechanisms to apply.

It is a phase that cannot be carried out for a set period of time but must be carried out for a long time to really take effect. It is not enough to implement the five previous phases, it is necessary to create a certain discipline so that the new modifications become part of the routine of the personnel involved, and therefore become easier and simpler.

During the implementation process of the commented phases, standardized work documents (labels, register lists, checklists, etc.) have been used to facilitate the registration and subsequent analysis of all the resources involved for decision-making. This documentation is relevant and key to the success of the implementation.

## 4. Results

If we want to have an impact on the organization, the workplace and the system efficiency, the Lean 6S methodology should not be understood as a one-off project.

Successful implementation of the 6S methodology begins when all members of the organization understand that 6S is a new way of working and, therefore, they must adapt their behavior, they must learn new things and they must make a continuous effort.

The 6S methodology has been applied in laboratories, classrooms and accesses and corridors: the total time of the trial laboratories implementation was 8 weeks (2 months), and the distribution is shown in Table 6.

The 6S must be prevented from becoming 3S, either due to the abuse of standardizing with visual control (SEIKETSU) that can lead to excessively autonomous control by the worker, or due to the abandonment of sustain with discipline and habit (SHITSUKE) so that it does not exist well-defined rules and responsibilities.

Based on these observations, which emerged during the implantation process, the results obtained in the indicated areas have been the following:A teamwork mentality was created that has increased the commitment of all participants, teachers, technical staff and students, including a better knowledge of the resources available in the laboratory.The time to disinfect resources for practices was reduced.More space was achieved in the work area, since unnecessary materials and supplies have been eliminated.No urgent cleaning and order processes have been needed.Resources are ordered and identified.Dirty sources have disappeared, and facilities are cleaned in less time.Faculty teachers and students can carry out a visual check, which allows to immediately detect deviations or errors.There is a commitment to maintaining previous results and continuous improvement.

One of the disadvantages is the necessary justification for the investment in health and safety resources when the project is being validated. It is advisable to use indicators that show the cost evolution in terms of operability during the implementation process. Some of these indicators can be customized over time to properly show what is happening in the facilities.

The used indicators were:Saturation degree in the entrances.Compliance with the entry and exit times.Compliance with standards of distancing and hygiene.Preparation time and adaptation of the faculty teacher’s job and laboratory practices under hygienic conditions.Caused errors by the incorrect use of the equipment (PPE).Maintenance activities.Anomalies identification time.Accidents infection rate.

Table 7 shows the first results obtained during the start of the 2020/2021 academic year with 2000 students.

Table 8 specifies the basic rules in the different areas of the School, while in Figure 6 and Figure 7 the results of some of the improvements obtained with the application of the Lean 6S methodology are shown through images.

## 5. Discussion

The good results provided in this work, which includes the implementation of a new Lean 6S methodology, show a new way of approaching the problem of the new coronavirus, without detriment to the educational quality required at the university levels of engineering schools. This new approach, motivated by the current special circumstances, is undoubtedly a step forward both in the fight against the current pandemic and in the approach of teaching in the field of engineering in situations of alarm or health emergency.

The demands caused by the epidemiological situation due to exposure to SARS-CoV-2 in educational centers, require standardized procedures to integrate and guarantee the safety and hygiene conditions of students, faculty members, and auxiliary service personnel.

In the engineering field of higher education centers, for the execution of theory teaching activities as well as practical laboratory classes, the required preventive measures must be combined with a commitment to teaching methodologies that ensure correct learning.

The application of lean methodologies in training organizations, as has been happening in other areas, such as health in hospitals and health centers [8,10,16], industrial [9,31,32], provides the basis for creating an organizational culture in situations such as that caused by SARS-CoV-2.

Usually, a work methodology in the industrial field must be implemented from the workers to the management. When the workers are well trained and the methodology is sound organized, implementation is usually not very difficult. Nevertheless, in the case of the university, students can be assimilated to workers in training, with the aggravation of youth, which has many advantages but discipline is not usually one of them.

The difficulty of maintaining the commitment to comply with the established rules by the entire group involved, especially the students, has been proven [33,34,35,36,37,38,39]. There is a clear tendency towards forgetting the required distancing and markers respect.

This work includes the application of this methodology in an engineering school physically located in Madrid, but it would be interesting to implement this methodology also in other engineering schools located in other environments, and in other faculties of other subjects. With the results of these new implementations of the methodology in more areas, there would surely be found specific problems different from those located here, the solution of which can improve this methodology.

As a new further line of work, a review of the methodology is proposed, which includes tools that ensure the commitment of the participants through mechanisms of motivation and involvement in the work environment, such as tools based on Design Thinking.

## 6. Conclusions

The Lean 6S methodology shown in this paper, based on the development of six phases, guarantees, thanks to the impact of all its phases and especially of three of them: cleaning (SEISO), standardize (SEIKETSU) and safety (SEKYURITI), the risk control of the health from SARS-CoV-2. This guarantee is achieved by permanently reviewing the safety of the educational center’s workstations.

The activities of sanitization, learning, control and maintenance of the involved resources are carried out in less time and with a considerable reduction in the caused cost, without forgetting the increase in the available space dedicated to the resource’s location.

The laboratories have been adapted to the security and organization conditions, that are required in the regulations required by the Occupational Risk Prevention Services against exposure to SARS-CoV-2. As indicated in the regulations, the appropriate protective equipment for the risk level is reviewed, the ordering of the workstations, the class attendance through the shifts organization, and the rearrangement of the common places where the maintenance of a minimum interpersonal safety distance between the faculty members, auxiliary services and students is guaranteed.

The effort of the faculty members in terms of following the established rules is notably increased. To balance this dedication, it is necessary to increase and rely on auxiliary personnel who guarantee rules compliance control in different spaces than the classroom and the laboratory.

When a group of technicians faces a problem, it may take more or less time to solve it, but once it is solved, this group of technicians is enriched and can be prepared to face the next problem with new and better tools. This is what has happened in the current situation. The first wave of the new coronavirus has certainly caught us off guard. We have had to improvise and mistakes have been made. However, the analysis of the situation and the study of the mistakes made is currently allowing us to face the second and future waves with better prospects, fewer improvisations and better results.

## Figures and Tables

**Figure 1 ijerph-17-09407-f001:**
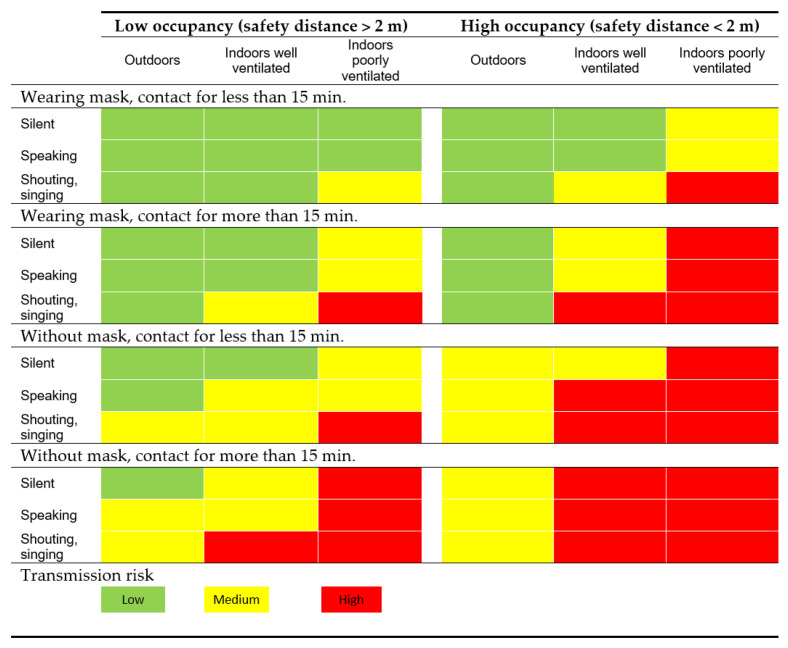
COVID-19 risk transmission in different situations.

**Figure 2 ijerph-17-09407-f002:**
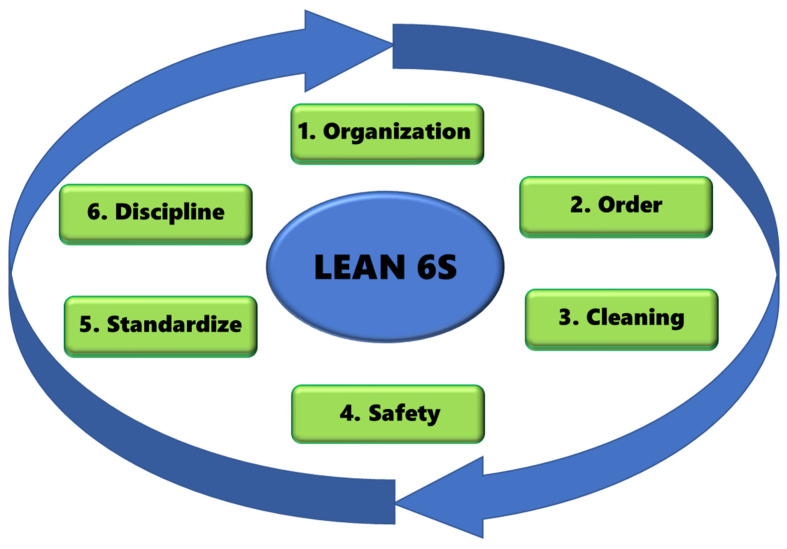
Lean 6S methodology.

**Figure 3 ijerph-17-09407-f003:**
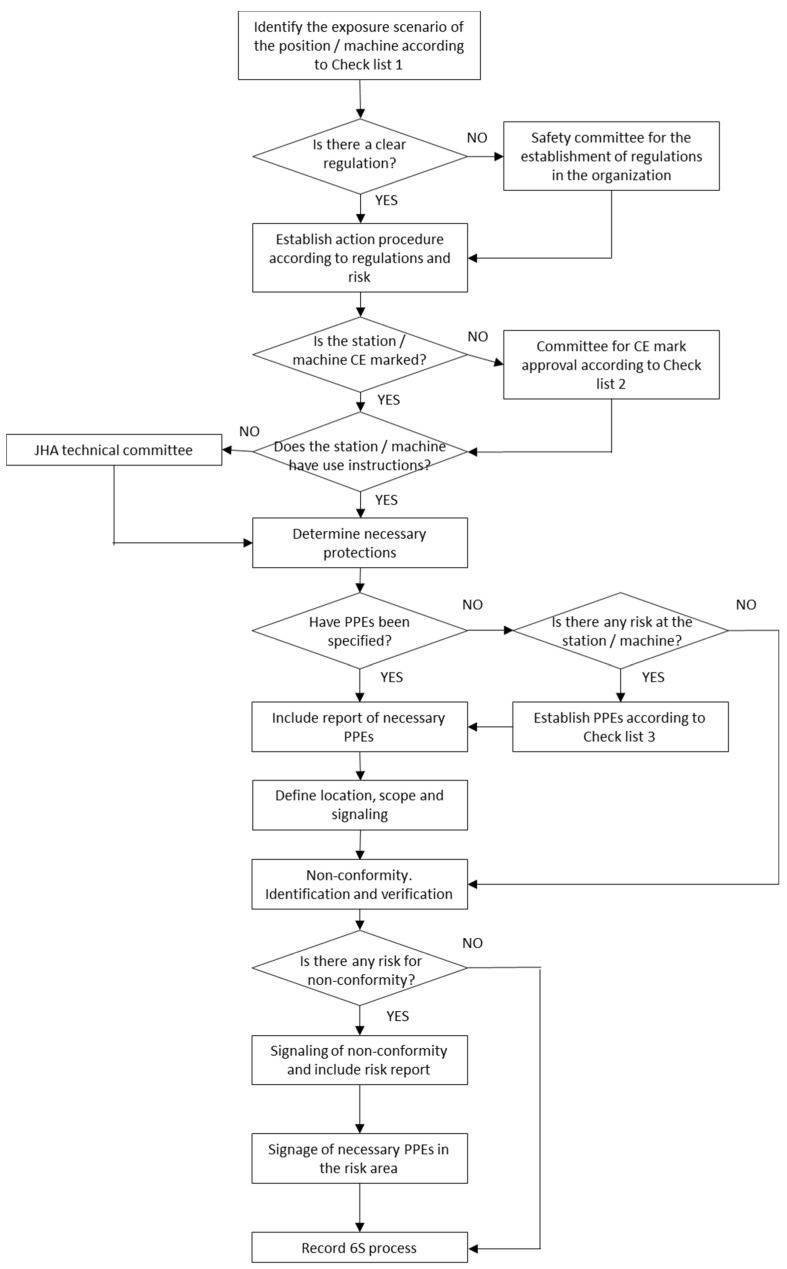
Safety implementation flow chart.

**Figure 4 ijerph-17-09407-f004:**
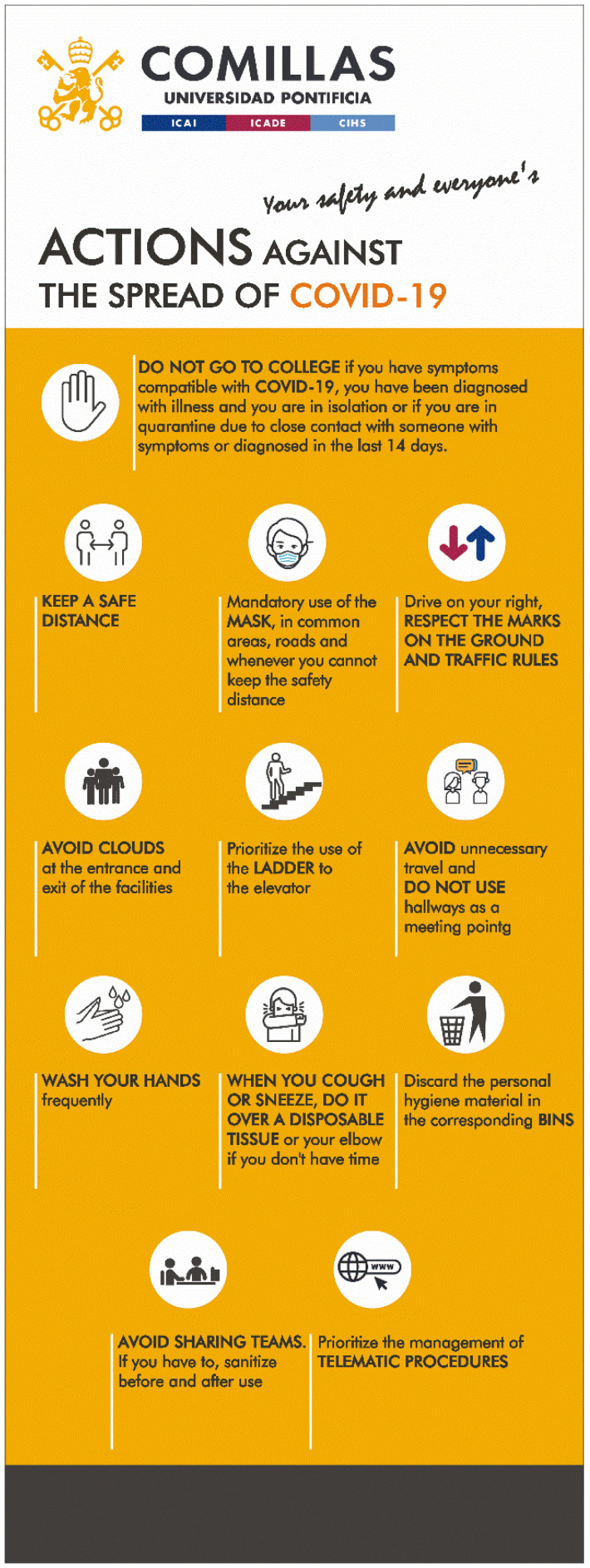
Informative markers.

**Figure 5 ijerph-17-09407-f005:**
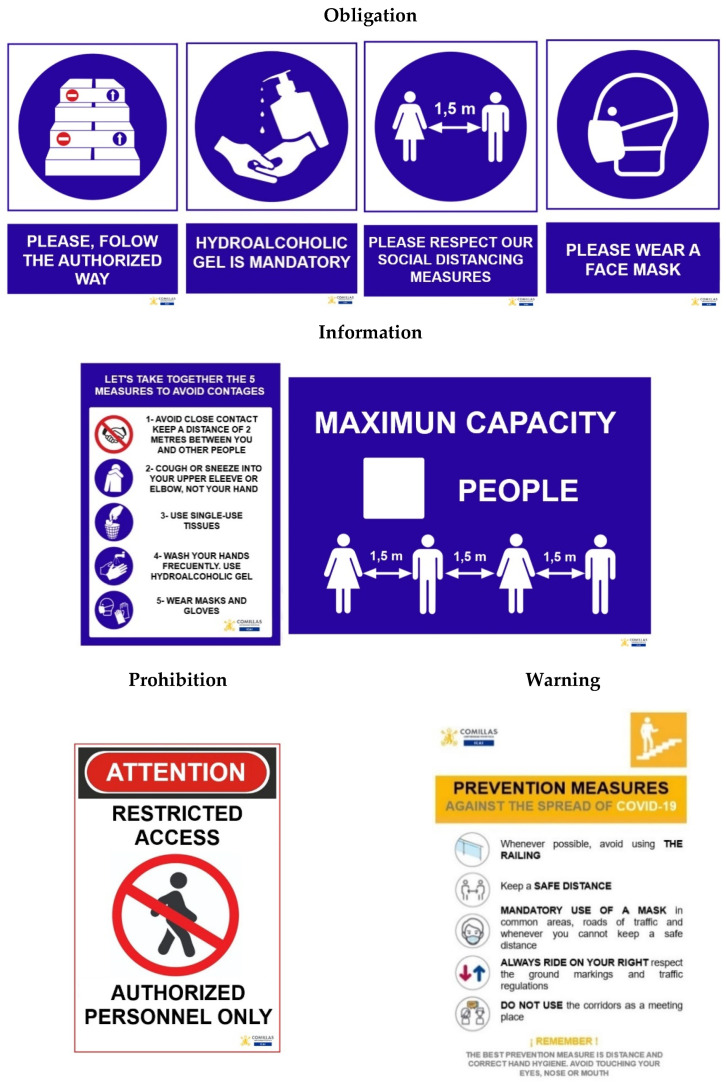
Informative markers.

**Figure 6 ijerph-17-09407-f006:**
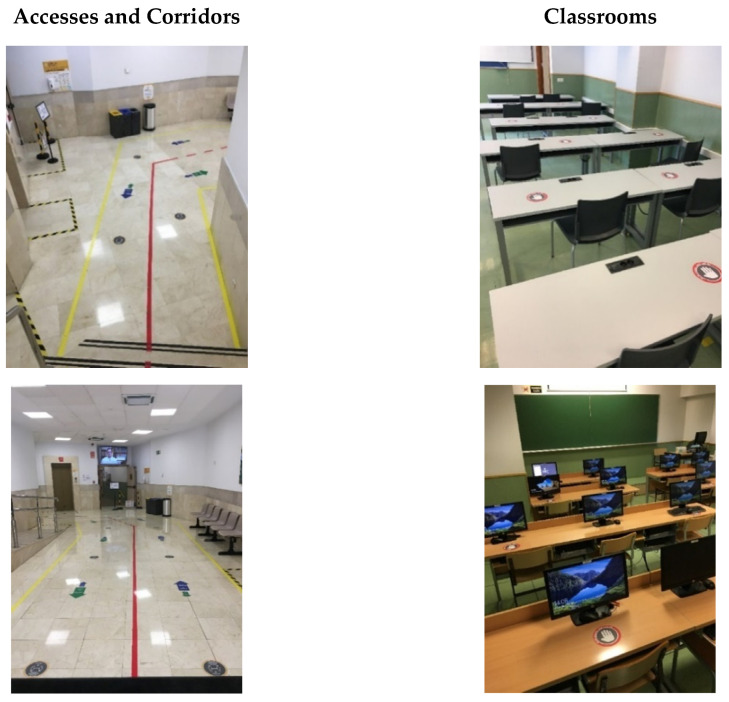
Visual comparison of entrances, classrooms, laboratories and other areas.

**Figure 7 ijerph-17-09407-f007:**
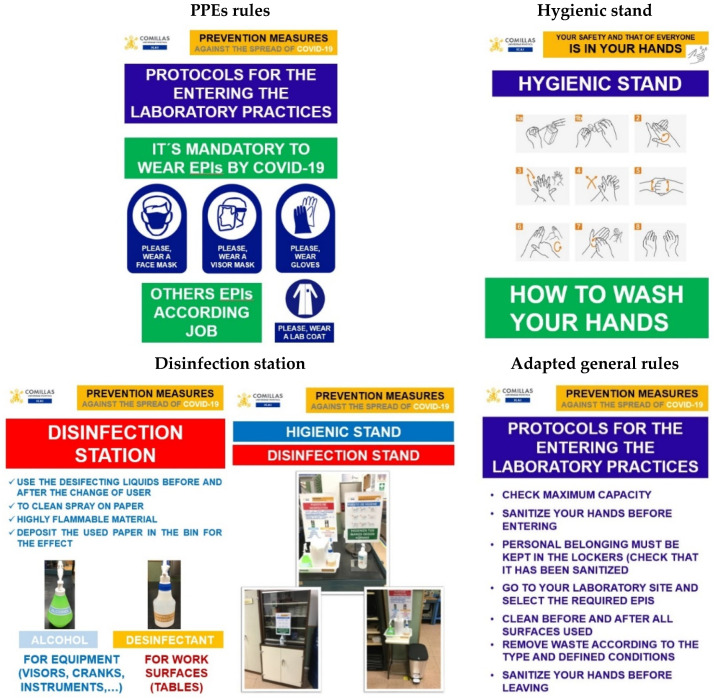
Adapted posters for carrying out the laboratory tests.

**Table 1 ijerph-17-09407-t001:** Exposure scenarios to SARS-CoV-2.

Scenario	Definition	Personnel
Risk exposure	Those work situations in which close contact with a suspected or confirmed case of SARS-CoV-2 infection may occur.	Healthcare and non-healthcare personnel attending a suspected or confirmed case of SARS-CoV-2. EXAMPLES: healthcare personnel from the educational center.
Low risk exposure	Those work situations in which the relationship that may have with a suspected or confirmed case does not include close contact.	Personnel whose work activity does not include close contact with a suspected or confirmed case of SARS-CoV-2. EXAMPLES: Teachers and staff of direct service to students, non-medical personnel who have contact with medical supplies, or contaminated waste.
Low probability of exposure	Workers who do not have direct attention to the public or, if they do, it occurs more than two meters distance, or they have collective protection measures that avoid contact (glass partition, etc.).	Personnel without direct attention to students, or more than 2 m distance, or with collective protection measures that avoid contact.EXAMPLES: Administrative staff, security personnel with a collective barrier.

**Table 2 ijerph-17-09407-t002:** 6S impact against SARS-CoV-2.

Phase	Impact
1. Organization (Seiri).	Classify, sorting. Identify and separate the necessary materials from the unnecessary ones to avoid a spread of the virus by contact with unnecessary resources.
2. Order (Seiton).	Setting an order of flow, streamlining. It establishes the way in which the necessary materials are identified and ordered. Avoid inadvertent contamination by clutter of resources.
3. Cleaning (Seiso).	Shining. Identify and eliminate dirty sources so that all resources are decontaminated, disinfected and if necessary sterilized.
4. Safety (Sekyuriti).	Supervise security. Ensures compliance with occupational safety regulations through the use of PPEs and compliance with cleaning and decontamination protocols.
5. Standardize (Seiketsu).	Visual control. Eliminate error by differentiating normal situations from abnormal or incorrect ones. It is achieved with visible and simple rules (signage).
6. Sustain (Shitsuke).	Discipline and habit. It forces to work and continuously review the established protocols (discipline).

**Table 3 ijerph-17-09407-t003:** Participants involved in the process implementation.

Direction	Formed by the Director, the Resources Deputy Director and the Department Director, being responsible for: The total responsibility for the 6S project. Ensuring commitment to maintenance and promotion of participation. Establishing the control process over the project implementation. Designating the operation area and the work team members.
Guide	This responsibility has been assumed by the head of the laboratory, as its main functions include the dynamics and the teamwork project coordination, executing the following actions: Training the team members in the 6S methodology. Collaborating with the directors in the implementation process planning. Ensuring the necessary resource availability. Ensuring the activities development, through team support and guidance. Keeping the 6S board indicators updated. Balancing the progress during the implantation process. Communicating results and experiences to other areas, facilitating the 6S methodology diffusion. Maintaining a continuous improvement of spirit in the know-how of the 6S methodology
Team	People involved in the implementation area: Two theory professors. Two lab professors. Two lab technical workers. Two students. The developed functions are: Training for 6S methodology. Project scheduling. Consulting the guide for people communication and training in the work area. Collecting and analyzing information, proposing ideas for improvement and seeking solutions with a teamwork approach. Tracking and analyzing the 6S board indicators.

**Table 4 ijerph-17-09407-t004:** Students information.

Number of Students	Degree	Student Demographics	Period	Other
1359	Degree	18–22 years	November 2019–October 2020	Madrid (Spain)
547	Master	22–26 years		
70	Master	26–45 years		

**Table 5 ijerph-17-09407-t005:** 6S process implementation.

Stage	Action	Recommendations
1	Management teamwork Training	→ prior awareness (rating other experiences) → detailed training on 6S → implementation guide reading → seeking potential expert support
2	Test area selection or review of areas implanted with Lean 5s previously	→ proper size → representative activity → stable, unchanging → representative → with receptive people → with improvement potential
3	Guide designation	→ resources director or laboratory manager → well trained in 6S → plan project capacity → form, encourage and recognize other → equipment users → manage meetings → seek materials support → edit and approve standard documents
4	Implementation team establishment	→ representative and multidisciplinary → 4/8 people → participation of different groups → participation of the director → minimum 40 man-hour dedication → initial training → tasks: quests, analysis, ideas, actions
5	Implementation planning	→ detailed planning → 2–4 months → provide time dedication and resources → budget preparation (recommended)
6	Launch meeting	→ with all the implementation team → only 6S general concepts → advantages to achieving the establishment → why implement it? → why this area? → why this team? → implementation plan
7	5S board establishment	→ involved team → before and after photos → establishment of process indicators → improvement plan in process
8	Implementation development	→ preparation → action, pictures, quests... → analysis and improvement plan → standardization
9	Results	→ in the end → communication to other people → feedback → learned lessons
10	Other laboratories implementation	→ go ahead taking into account criteria of → the pilot laboratory → take advantage of the acquired knowhow → take advantage of the initial team → support
11	Continuous improvement	→ periodical review → indicators monitoring → further training and learning → suggestions → advanced courses → experiences interchange forums

Errors to avoid: lack of commitment to direction, insufficient time dedicated, newly incorporated guide inexpert, skipped methodology steps, selecting a large or not representative experimental lab, thinking that the project ends in the 6S.

**Table 6 ijerph-17-09407-t006:** Time expected of 6S methodology implementation.

Area	Guide	Teamwork	Personal Area
Laboratories	6 h/phase	4 h/phase	- organization: 3 h - order: 3 h - cleaning: 2 h - training: 1 h/phase
Classroom	4 h/phase	4 h/phase	- organization: 3 h - order: 3 h - cleaning: 2 h - training: 1 h/phase
Accesses and corridors	4 h/phase	4 h/phase	- organization: 10 h - order: 5 h - cleaning: 6 h - training: 2 h/phase

**Table 7 ijerph-17-09407-t007:** Implantation results.

Indicator	Compliance Degree	Observations
Entrances saturation control	100 %	With three accesses and control by mandatory visual identification (bracelet).
Entry and exit times compliance	100%	Two entry times: 7:50 a.m. to 8:00 a.m. and 8 to 8:10 a.m. Two departure times: 11:50 a.m. to 12:00 p.m. and from 12:00 to 12:10 h.
Distancing and hygiene rules compliance	97.5%	A greater compliance commitment is required, mainly during rest hours between activities and intermediate trips.
Compliance with the preparation time and adaptation of the work areas: classes, laboratories.	95%	Prior sanitization and disinfection, PPEs verification and synchronization of video and audio resources are required.
Errors in the incorrect use of equipment in laboratories test.	98.5%	Prior knowledge and location of the PPEs recommended is required. In some laboratory test, e.g., welding systems, and tool-machines, relaxes the use of goggles and visors.
Preventive and corrective maintenance activities for the safety of equipment and work areas	90%	There is an increase in resource replenishment and permanent review of safety and hygiene conditions that require the review of maintenance plans.
Anomaly identification time	100%	Visual impact ensures immediate identification.
Infection rate due to internal contact	0%	The PPEs used are guaranteeing a 0 rate in teaching activities.
**Indicator**	**Increase relative to the value under normal conditions**	**Observations**
Laboratory practice preparation time	>15%	It is required to ensure disinfection before and after the use of resources.
Theory class preparation time.	>10%	It is required to ensure the correct use of PPEs. More time is needed to synchronize digital resources for combined face-to-face and remote classes (bimodal system).
Time of access to the job place.	>10%	The safety distance and accumulation of people in accesses must be modified access times.
Risk situations due to improper use of protective equipment (PPE).	>5%	In some laboratory practices, e.g., welding systems, and tool-machines, relaxes the use of goggles and visors.
Maintenance and replacement costs.	>10%	Higher cost and time are required to replace consumables and check PPEs.
**Indicator**	**Number of Anomalies**	**Observations**
Identification of security anomalies	3	Safety distance reduction during the examination period due to errors in the access control procedure. Errors in mobility through internal areas, due to the fact that some visual safety indications do not provoke their demand due to their lack of impact (design and location). Relaxation of the safety distance during rest intervals between classes and in internal mobility.

**Table 8 ijerph-17-09407-t008:** Basic rules in the School.

**Access Rules**	**Classroom Rules**
- Access through the assigned door at the assigned time (shift) and identify yourself with the colored bracelet provided. - If the entrance hall is full, queue in the indicated direction. - Once in the hall, at the control point, sanitize your hands with the hydroalcoholic gel dispensers. - Go up the ladder to your right, respecting the safety distance and avoiding touching the railing. - When you get to your floor, access the corridor on your right.	- Sit in one of the designated positions, which you will not change during the assigned shift. - Do not remove your mask in the classroom. - Avoid sharing objects or devices in class and in any other space of the faculty.
**Laboratory Rules**	**Break Rules**
- Check the capacity allowed. - Before entering, sanitize your hands. - Leave your belongings in the assigned locker (check that it was sanitized). - Go to your internship position and select the required PPE. - Cleans used resources and surfaces before and after. - Remove waste according to the type and conditions defined. - Before leaving, sanitize your hands.	- Minimize displacements. - You can leave the classroom to go to the assigned toilet, using the corridor circulating to the right. - Identify the available capacity in the toilet. - Sanitize your hands before and after using the toilet - You can use the tap of the water source (never the spout) with a bottle and clean the handle with hydroalcoholic gel once used.
**Exit Rules**	
- Access the hallway and use the assigned ladder, always moving to your right and maintaining a safe distance. - Sanitize your hands in any of the dispensers. - Very important, do not stop when leaving the building because you will block the evacuation of the rest of your colleagues. - If you want to smoke, you will have to separate yourself at least 2 m from the entrance access and respect the safety distance.	

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
