# Peer review of "Application of Lean 6s Methodology in an Engineering Education Environment during the SARS-CoV-2 Pandemic"

_ijerph, 2020, doi:10.3390/ijerph17249407_

Round 1

Reviewer 1 Report

For this point in time when global epidemics are at risk, research such as this one would be useful for global epidemic preparedness! In this study, the authors use very practical process phase definitions as well as recommendations that will really inform the reader of what is available. However, this article would be more useful to the reader in practical research if it were revised with reference to the following suggestions:

1. Define reference indicators for input and output at various stages:
There are reference indicators and workflow diagrams provided for each stage of the study that will be helpful to the reader! But would it be more helpful to the reader to add advice on risk at all stages! For example, different proposals may be implemented at different stages, but are there other implications that may arise from the implementation of different proposals? What is the author's advice at this point?

2. Please use consistent terminology and explain important key words. For example, the title in figure 1 is proposed to be modified to be consistent (Organization) and a brief description is added (line 131). For terms that appear frequently in the text (e.g., EPIs), please provide the full name and description of the term the first time it appears, so that it is clear to the reader and does not cause confusion with other terms.

The following two suggestions could be added to the 4. Discussion and conclusions to make the study more useful to the readers! The proposed study separates the discussion from the conclusion! Include in the discussion issues that arise during the course of the project, including those that have been resolved. So that future researchers involved can avoid mistakes, or how they could be better! In the conclusion, can we ask the researchers to provide some theoretical and methodological corrections or additions to the suggestions made during the implementation of Lean 6s?

3. Could the proposed study add some similar success stories using Lean 6s or similar cases from other industries to support the theoretical approach?

4. The joy of successful improvement with this successful project is worth sharing! Are there some specific recommendations that are suggested to be provided to readers so that they can share their experiences from this study to avoid the project management problems that can easily arise? For example, the need to obtain authorization from some organizations to implement a project to avoid creating conflict issues with relevant stakeholders! Or other issues that actually occurred during the course of the project but were properly resolved? How were they solved in the process, these experiences deserve to be shared!

The biggest difference between this and the results is the point in time at which the recommendations in the results can provide readers with solutions to similar problems in the course of implementation. And the suggestions here are so that the reader can technically avoid these problems during project planning!

Author Response

Dear Reviewer,

We greatly appreciate your thoughtful comments that helped improve the manuscript. We trust that all of your comments have been addressed accordingly in a revised manuscript. Thank you very much for your effort.

Comments and Suggestions for Authors

For this point in time when global epidemics are at risk, research such as this one would be useful for global epidemic preparedness! In this study, the authors use very practical process phase definitions as well as recommendations that will really inform the reader of what is available. However, this article would be more useful to the reader in practical research if it were revised with reference to the following suggestions:

  1. Define reference indicators for input and output at various stages: We have reviewed this issue.

There are reference indicators and workflow diagrams provided for each stage of the study that will be helpful to the reader! But would it be more helpful to the reader to add advice on risk at all stages! For example, different proposals may be implemented at different stages, but are there other implications that may arise from the implementation of different proposals? What is the author's advice at this point?

  1. Please use consistent terminology and explain important key words. For example, the title in figure 1 is proposed to be modified to be consistent (Organization) and a brief description is added (line 131). For terms that appear frequently in the text (e.g., EPIs), please provide the full name and description of the term the first time it appears, so that it is clear to the reader and does not cause confusion with other terms. We have checked this issue.

The following two suggestions could be added to the 4. Discussion and conclusions to make the study more useful to the readers! The proposed study separates the discussion from the conclusion! Include in the discussion issues that arise during the course of the project, including those that have been resolved. So that future researchers involved can avoid mistakes, or how they could be better! In the conclusion, can we ask the researchers to provide some theoretical and methodological corrections or additions to the suggestions made during the implementation of Lean 6s? We have separated and implemented the sections Discussion and Conclusions.

  1. Could the proposed study add some similar success stories using Lean 6s or similar cases from other industries to support the theoretical approach? We have added some new references in this sense.
  2. The joy of successful improvement with this successful project is worth sharing! Are there some specific recommendations that are suggested to be provided to readers so that they can share their experiences from this study to avoid the project management problems that can easily arise? For example, the need to obtain authorization from some organizations to implement a project to avoid creating conflict issues with relevant stakeholders! Or other issues that actually occurred during the course of the project but were properly resolved? How were they solved in the process; these experiences deserve to be shared! We have included some specific recommendations.

The biggest difference between this and the results is the point in time at which the recommendations in the results can provide readers with solutions to similar problems in the course of implementation. And the suggestions here are so that the reader can technically avoid these problems during project planning! We have added some recommendations.

Reviewer 2 Report

The prevention of SARS-CoV-19 is an urgent issue facing the whole world and it is important to accumulate knowledge in countries and regions where the epidemic is threatened. Baesd on this understanding, in order to be accepted as an article, the following points must be intensively addressed.

First, the introduction should explain the issues that are the premise of this study then research questions. Specifically, the region and environment covered by this study, the state of anti-infectious management from the past to the present, the current status and challenges of SARS-CoV-2 countermeasures, and the reasons for the selection of the present methodology/approach should be mentioned.

Secondly, LEAN and 5S/6S, which are key concepts in this study, need more elaborate explanation in L56-62. After organizing these concepts based on the genealogy of previous studies, the significance, utility and challenges of each S must be discussed. In fact, the authors interpreted the 6Ss as seiri, seiton, seiso, safety, seiketsu and shitsuke (Table 2), but the actual 6Ss are seiri, seiton, seiso, seiketsu, 'sahou' and shitsuke/shukan.

Third, while this study takes the form of an empirical study, it completely lacks the description of the methodology. There should be a separate chapter after the introduction, which should specifically describe what observations were made and what procedures were followed to conduct this study.

Fourth, in relation to the above point, the content of the description in the Result section remains to a narrative of the authors' opinions and cannot be described as the result of a scientific research. Specific and analytical results need to be clearly presented based on each of observed facts in the comparison to a control.

Finally, the novelty of the study should be clearly explained in the Discussion section, and the limitations and prospects of the study should also be discussed.

The following are minor points that need to be addressed as well:

Explanations are needed as to why seiri, seiton and shitsuke were excluded from the scope of this study despite adopting the 6S framework.

L96-125 explains the main points and procedures to be adhered to, but what norms are they based on? The background of the proposal should be carefully explained based on the source so that it does not appear to be merely the author's idea.

The legend / caption is absent in each of these figures.

Figure 1 is an important assumption in this study, but its robustness is questionable. Although this figure is referred to as a citation of reference [20], it needs to be verified and shown when and in which areas the observations were made and whether the results are truly valid.

The quality of Figure 4 needs to be improved. The handwritten text at the top should be eliminated.

It needs to be written faithfully to the template set by the journal: newlines/paragraphs, fonts and the defined format of references, etc.

The present study is supposed to be empirical by enrolling human subjects and/or actual workers. In that case, a proof of an ethical review result conducted by the research institute should be provided as 'ethical considerations' in the very last of the manuscript.

Author Response

Dear Reviewer,

We greatly appreciate your thoughtful comments that helped improve the manuscript. We trust that all of your comments have been addressed accordingly in a revised manuscript. Thank you very much for your effort.

Comments and Suggestions for Authors

The prevention of SARS-CoV-19 is an urgent issue facing the whole world and it is important to accumulate knowledge in countries and regions where the epidemic is threatened. Based on this understanding, in order to be accepted as an article, the following points must be intensively addressed.

First, the introduction should explain the issues that are the premise of this study then research questions. Specifically, the region and environment covered by this study, the state of anti-infectious management from the past to the present, the current status and challenges of SARS-CoV-2 countermeasures, and the reasons for the selection of the present methodology/approach should be mentioned. We have reviewed and modified this section.

Secondly, LEAN and 5S/6S, which are key concepts in this study, need more elaborate explanation in L56-62. After organizing these concepts based on the genealogy of previous studies, the significance, utility and challenges of each S must be discussed. In fact, the authors interpreted the 6Ss as seiri, seiton, seiso, safety, seiketsu and shitsuke (Table 2), but the actual 6Ss are seiri, seiton, seiso, seiketsu, 'sahou' and shitsuke/shukan. This is our interpretation based on the bibliography (see Jiménez et al., 2019).

Third, while this study takes the form of an empirical study, it completely lacks the description of the methodology. There should be a separate chapter after the introduction, which should specifically describe what observations were made and what procedures were followed to conduct this study. We have modified the text in order to include this recommendation and we have reorganized sections 2 and 3.

Fourth, in relation to the above point, the content of the description in the Result section remains to a narrative of the authors' opinions and cannot be described as the result of a scientific research. Specific and analytical results need to be clearly presented based on each of observed facts in the comparison to a control. We have changed this section.

Finally, the novelty of the study should be clearly explained in the Discussion section, and the limitations and prospects of the study should also be discussed. We have separated and implemented Discussion and Conclusions.

The following are minor points that need to be addressed as well: Checked.

Explanations are needed as to why seiri, seiton and shitsuke were excluded from the scope of this study despite adopting the 6S framework.

L96-125 explains the main points and procedures to be adhered to, but what norms are they based on? The background of the proposal should be carefully explained based on the source so that it does not appear to be merely the author's idea.

The legend / caption is absent in each of these figures.

Figure 1 is an important assumption in this study, but its robustness is questionable. Although this figure is referred to as a citation of reference [20], it needs to be verified and shown when and in which areas the observations were made and whether the results are truly valid.

The quality of Figure 4 needs to be improved. The handwritten text at the top should be eliminated.

It needs to be written faithfully to the template set by the journal: newlines/paragraphs, fonts and the defined format of references, etc.

The present study is supposed to be empirical by enrolling human subjects and/or actual workers. In that case, a proof of an ethical review result conducted by the research institute should be provided as 'ethical considerations' in the very last of the manuscript.

Reviewer 3 Report

Minor remarks:

  • Why lean is mostly capitalised, but sometimes it just starts with capital letter? I do not see the need to capitalise it at all.
  • "Government organizations responsible of health management have generated "Action procedures for 40 Occupational Risk Prevention Services against exposure to SARS-CoV-2" [1]." - This is true for Spain, but no eveidence is given for otyher countries and have not declared, that the work is limited just to Spain. What about other EU countries, US, UK, Australia, China, Japan? Are there any practices worth to be discussed? If not, you should state it clearly and prove.
  • "the new coronavirus" - I suggest to use exact name.
  • lines 44-48 - And what about action plans for recurring waves and recurring lockdowns and moving all education again to remote mode? Please comment on this issue.
  • lines 56-57 - lean means lean, and agile has different focus, you have to be lean to be agile, but not every lean organization becomes agile by default. I think here is shortcut, which you should avoid.
  • Line 61 - You named 6th S before listing all the original 5S.
  • Consider if 6S is really methodology or rather it is better to call it tool or technique?
  • In section 2 you mixed some parts which in fact related to introduction or state-of-the-art. I do not also see the methodology of your research being depicted. I suugest to illustrate it on figure and show the logic of your research process. You also described here some results of your work (e.g. what action should be taken within each S), but I do not see any methodological explanation how have you approached to your findings. It would be good to formulate hypothesis and research questions for your work, and show how to verify and answer them in your methodology. How can you prove that actions which you propose for each S are sufficient and you have not ommited anything important? Such issues should be explained in your methodology. Also, you should show which research methods have you applied and why? You use case study, but you need to elaborate more (why? which type? what is the goal?, etc).
  • Reorganize section 2 to make to purely focused on methodology. Extend methodology description as discussed above. Extract parts of section 2 to Section 1 and Section 3. Your recommendation on specifity of each S is in fact result of your research, and then in Section 3 you described verification of your 6S tailored for COVID and engineering education environment.
  • Fig. 4 - Definitely too low quality.
  • Discussion is too general. There are no limitations discussed extensively, nor further research. Section 4 contains mainly conclusions, but not discussion.

This is very interesting topic and timely. It also fits well to the journal scope. However, the paper misses scintific rigour and I suggest to include my remarks listed above.

Author Response

Dear Reviewer,

We greatly appreciate your thoughtful comments that helped improve the manuscript. We trust that all of your comments have been addressed accordingly in a revised manuscript. Thank you very much for your effort.

Comments and Suggestions for Authors

Minor remarks:

Why lean is mostly capitalised, but sometimes it just starts with capital letter? I do not see the need to capitalise it at all. Checked.

"Government organizations responsible of health management have generated "Action procedures for 40 Occupational Risk Prevention Services against exposure to SARS-CoV-2" [1]." - This is true for Spain, but no eveidence is given for otyher countries and have not declared, that the work is limited just to Spain. What about other EU countries, US, UK, Australia, China, Japan? Are there any practices worth to be discussed? If not, you should state it clearly and prove. We have clarified this point.

"the new coronavirus" - I suggest to use exact name. Checked.

lines 44-48 - And what about action plans for recurring waves and recurring lockdowns and moving all education again to remote mode? Please comment on this issue. We have explained this issue.

lines 56-57 - lean means lean, and agile has different focus, you have to be lean to be agile, but not every lean organization becomes agile by default. I think here is shortcut, which you should avoid. Checked.

Line 61 - You named 6th S before listing all the original 5S. We have previously listed the original 5S.

Consider if 6S is really methodology or rather it is better to call it tool or technique? It could be possible to call it tool but we have used it as a methodology based on the bibliography.

In section 2 you mixed some parts which in fact related to introduction or state-of-the-art. I do not also see the methodology of your research being depicted. I suugest to illustrate it on figure and show the logic of your research process. You also described here some results of your work (e.g. what action should be taken within each S), but I do not see any methodological explanation how have you approached to your findings. It would be good to formulate hypothesis and research questions for your work, and show how to verify and answer them in your methodology. How can you prove that actions which you propose for each S are sufficient and you have not ommited anything important? Such issues should be explained in your methodology. Also, you should show which research methods have you applied and why? You use case study, but you need to elaborate more (why? which type? what is the goal?, etc). Checked.

Reorganize section 2 to make to purely focused on methodology. Extend methodology description as discussed above. Extract parts of section 2 to Section 1 and Section 3. Your recommendation on specifity of each S is in fact result of your research, and then in Section 3 you described verification of your 6S tailored for COVID and engineering education environment. We have reorganized sections 2 and 3.

Fig. 4 - Definitely too low quality. We have modified this figure.

Discussion is too general. There are no limitations discussed extensively, nor further research. Section 4 contains mainly conclusions, but not discussion. We have separated and implemented the sections Discussion and Conclusions.

This is very interesting topic and timely. It also fits well to the journal scope. However, the paper misses scintific rigour and I suggest to include my remarks listed above. Thank you very much for your recommendations.

Round 2

Reviewer 2 Report

I confirm that appropriate actions have been taken to address the requested points for improvement, but one on the third point for the method. Please describe things of the subjects in the present study, i.e., the number (and demographics) of the participants, the enrolment process, and the period of the empirical research, etc.

Author Response

We ha added this table, thank you for the recommendation

Number of students

Degree

Student demographics

Period

Other

1359

Degree

18-22 years

november 2019 – october 2020

Madrid (Spain)

547

Master

22-26 years

70

Master

26-45 years